# Enhancing the gravity model for commuters: Time-and-spatial-structure-based improvements in Japan's metropolitan areas

Yixuan Y. Zheng[1], Yohei Shida[2,3], Hideki Takayasu[3], Misako Takayasu[1,3]*

**1** Department of Systems and Control Engineering, School of Engineering, Institute of Science Tokyo, Yokohama, Kanagawa, Japan, **2** Institute of Systems and Information Engineering, University of Tsukuba, Tsukuba, Ibaraki, Japan, **3** Department of Computer Science, School of Computing, Institute of Science Tokyo, Yokohama, Kanagawa, Japan

* takayasu@comp.isct.ac.jp

**Data availability statement:** Data cannot be shared publicly because data is available only

## Abstract

Metropolitan commuting flows reveal crucial insights into urban spatial dynamics; however, existing mobility models often struggle to capture the complex, heterogeneous patterns within these regions. This study introduces the Spatially Segregated Urban Gravity (SSUG) model, a novel approach that synergistically combines urban classification with gravity-based flow prediction to address this limitation.

The SSUG model's key innovations include: (1) demonstrating the existence of different scaling laws in metropolitan areas, (2) identifying the existence of data-driven bifurcation that delineates urban-suburban commuting behaviors, (3) employing scaling exponents to reveal spatial segregation, and (4) leveraging high-resolution Global Positioning System (GPS) data for precise deterrence factor measurement. This multifaceted approach enables simultaneous improvement in flow prediction accuracy and robust urban functional classification.

Empirical validation across six diverse Japanese metropolitan areas—Tokyo, Osaka, Nagoya, Fukuoka, Sendai, and Sapporo—demonstrates the SSUG model's superior predictive power compared to traditional gravity models. Our results unveil previously undetected patterns of spatial structure and functional segregation, particularly highlighting the divergent commuting dynamics between urban cores and suburban peripheries.

The SSUG model's capacity to discern fine-grained urban-suburban differences while accurately forecasting commuting flows offers transformative potential for evidence-based urban planning. By providing a more nuanced understanding of metropolitan mobility patterns, this study equips policymakers with a powerful tool for optimizing transportation networks, refining land-use strategies, and fostering sustainable urban development in increasingly complex metropolitan landscapes.

on request from a third party. Data are available from Agoop Corporation, a Japanese private company that provides location information big data acquired from smartphone applications. The specific product is "Pointo-gata ryoudou-jinkou data" (Point-type population data). Interested researchers who meet the criteria for access to confidential data can visit https://agoop.co.jp/service/dynamic-population-data/ for more information.

**Funding:** This work was supported by the Japan Society for the Promotion of Science, Grant-in-Aid for Scientific Research (B) (Grant Number 23K22980 to MT). The funder had no role in study design, data collection and analysis, decision to publish, or preparation of the manuscript.

**Competing interests:** The authors have declared that no competing interests exist.

# 1 Introduction

## 1.1 Background

Human mobility patterns are fundamental to understanding urban structure and function, playing a crucial role in shaping how cities grow, operate, and evolve. Accurate modeling of these patterns is essential for effective urban planning, transportation design, and resource allocation [1,2]. As metropolitan areas continue to expand, the ability to predict and manage mobility patterns becomes increasingly vital for addressing major urban challenges, including traffic congestion, public health crises, and access to jobs and services [3,4].

Metropolitan areas, characterized by dense populations and complex spatial dynamics, require a more nuanced approach to mobility analysis compared to intercity travel. Intracity mobility in these areas involves shorter, more frequent trips influenced by factors such as population density, land use, and public transportation availability [5–7]. This complexity is reflected in the hierarchical organization of urban mobility, where trip flows are closely linked to urban livability and environmental indicators [8,9].

Various models have been developed to capture different mobility characteristics, each with its strengths and limitations. The radiation model, while effective for intercity flows, performs poorly at the city scale [10]. The population-weighted opportunities (PWO) model and rank-based models better account for the hierarchical nature of urban mobility [11]. The gravity model remains widely used, positing that the interaction between locations is directly proportional to their populations and inversely proportional to the distance between them [12].

Within metropolitan regions, commuting flows offer particularly valuable insights into the interaction between residential areas and workplaces. These flows reveal spatial dynamics crucial for optimizing transportation systems and addressing urban inequalities [4,13]. The quantitative characteristics of commuting mobility also play a pivotal role in classifying urban functions, offering key insights into socio-economic dynamics within urban areas [14–16].

## 1.2 Problem statement

Despite advancements in mobility modeling, two critical challenges persist in estimating intracity commuting flows:

1. Precisely measuring deterrence factors and driving forces: Existing studies have enhanced models by modifying deterrence factors' scaling forms [17], refining measurement variables [18], and incorporating cost functions [8,11]. Driving forces have seen improvements through proxies like the extent of human activity [8,19].
2. Appropriately configuring models for different urban contexts: Different demographic groups and regions with varying urban structures require distinct model adaptations [5,15,16]. However, classification often presents challenges in establishing grouping criteria, as deeper functional differences within regions can be elusive.

Current methods for classifying urban regions often rely on predetermined administrative boundaries [5,20] or arbitrary population density thresholds [15,16], such as Japan's Densely Inhabited Districts (DID) data [21,22]. These approaches frequently neglect to account for the distinct functions of areas with similar population densities, such as residential versus workplace zones, which play unique roles in shaping urban dynamics [4,17,23,24].

There remains a significant gap in developing models that can effectively analyze commuting patterns across different metropolitan regions, particularly in understanding the interactions between residential and workplace distributions. This limitation hinders our ability

to accurately capture mobility patterns and understand functional differences in complex metropolitan areas.

## 1.3 Research purpose and methodology

While recent advances in deep learning have enabled gravity models to automatically adapt to unfamiliar metropolitan regions [25] by learning numerous complex urban life patterns, this research emphasizes the importance of a model's regional adaptability. The key lies in the model's ability to accurately capture identical human mobility patterns across different regions. To achieve this regional adaptability, understanding the basic rhythm of city life becomes crucial, particularly the commuting flows, that are primarily shaped by the spatial distribution of residential and workplace locations. This fundamental pattern serves as the cornerstone for modeling human mobility across different metropolitan areas.

To address these challenges, this study introduces the Spatially Segregated Urban Gravity (SSUG) model, an innovative approach that integrates urban classification within the mobility modeling framework. The SSUG model not only refines human mobility flow predictions but also uniquely distinguishes between residential and workplace zones to capture the distinct commuting dynamics of urban and suburban areas within metropolitan regions.

Key innovations of the SSUG model include:

- Identification of a statistically significant bifurcation in commuting patterns, integrated as a model parameter.
- Implementation of distinct criteria for residential and workplace zones to reflect their unique dynamics.
- Application of origin and destination scaling exponents to enhance spatial segregation analysis in both residential and workplace contexts.
- Utilization of high-accuracy GPS data with 1-minute intervals for measuring commuting time in metropolitan areas.

Our methodological approach begins by identifying the key point where commuting patterns change, confirmed through rigorous statistical tests. This evidence of non-single parameters suggests a natural segregation point within the data, which is then incorporated into the gravity model to achieve a more accurate representation of commuting flows.

The SSUG model offers several advantages over traditional gravity models:

- Integration of urban classification within the gravity model framework, providing dual functionality in flow prediction and urban classification.
- A data-driven approach that closely aligns with real-world commuting behaviors, allowing for clear differentiation of patterns among functional areas.
- A comprehensive understanding of urban dynamics by considering the unique characteristics of residential and workplace zones.

To validate the model's effectiveness, we applied it to six major metropolitan areas in Japan, to test if the model learns the identical city patterns or not [25], and demonstrating superior performance in predicting commuting flows compared to traditional gravity models.

This study holds significant implications for urban planners and policymakers, offering a more nuanced and accurate view of commuting flows to inform transportation planning, land-use optimization, and sustainable development initiatives in metropolitan regions.

The remainder of this paper is structured as follows, Sects 2 and 3 detail the methodology and data used in developing the SSUG model. Sect 4 presents the results of our model

application to Japanese metropolitan areas and discusses the implications of our findings and potential applications. Finally, Sect 5 concludes the study and suggests directions for future research.

## 2 Study area and data

### 2.1 Study area

We selected six of the most populated metropolitan areas in Japan—Tokyo, Osaka, Nagoya, Fukuoka, Sapporo, and Sendai (S1 Fig (S1 File)) - each with a population exceeding 2 million, according to the latest official data [26]. These areas were chosen based on their economic development and population scale, making them representative of urban commuting behavior in Japan.

For this study, each metropolitan area was divided into 1 km × 1 km grid cells. In the case of the Tokyo metropolitan area, this resulted in approximately 13,000 grid cells.

### 2.2 Data description

The mobility GPS dataset used in this study was provided by a Japanese company, Agoop Corp. The dataset includes anonymized location data from approximately 1 million smartphone users per day across Japan. The data points include user ID, timestamp, coordinates, and home and work city codes, with an average location accuracy of about 10 meters.

The study focuses on weekdays throughout 2021, excluding weekends and holidays, to concentrate on typical workday commuting patterns. This exclusion allows for a more precise analysis of regular commuting behaviors, though it also means that the study does not account for variations in mobility that occur during weekends and holidays. As a result, the findings may be less applicable to understanding non-work-related travel patterns, which could have different implications for urban planning and transportation policies. To test the robustness of the SSUG model under varying conditions, we also applied the same approach to 2020 data, which was significantly impacted by the COVID-19 pandemic, and the analysis of 2020 data yielded results that were largely consistent with our 2021 findings.

The dataset is limited to GPS information from smartphones with applications issued by Agoop Corp., introducing some potential biases. According to the Ministry of Internal Affairs and Communications, Japan, there is an age bias in smartphone usage, leading to the under-representation of elderly individuals and children under 13 in the data. This could result in a skewed analysis that primarily reflects the commuting patterns of working-age adults.

To ensure privacy, user IDs were randomized each night, making it impossible to track an individual's movement across multiple days. Additionally, the precise residential locations of users are not available; the data is blurred to a central point within a 1 km-square or, in less populated areas, a 10 km-square mesh. However, the dataset does include the names of the resident city and workplace city at the municipality level.

We applied several filtering criteria to the data to focus on commuting behavior: 1) only users with more than 100 points per day were included, and 2) we selected IDs where both the home city code and work city code were within the target areas. These criteria ensured that the dataset provided sufficient geolocation information to accurately reflect each user's commuting patterns. The final dataset includes approximately 10 million anonymized smartphone users per day across the six study areas, with an average of 125 trajectory points per ID per weekday.

We counted the number of GPS users at 5:00 each day to estimate the home prefecture population and calculated the effective population ratio by comparing the actual population

of each prefecture (as of October 2021) [27] to the number of GPS users recorded, and the results are shown in Sect 9 (S1 File). This ratio was then used to renormalize the GPS data points, aligning them more closely with the actual population distribution. By applying this renormalization process, we ensured that the GPS dataset provided by Agoop Corp. accurately reflects the true population figures, thereby enhancing the representativeness of the data.

# 3 Method

## 3.1 Data processing

To analyze the commuting mobility dataset, this study first identified each individual's home and work locations from the raw trajectory data, allowing us to extract commuter user IDs from the mass of GPS data. Without survey data, and relying on high-accuracy trajectory data with a 1-minute time interval, this study estimated commuting times based on precisely identified home and work locations. The methods for identifying home and work and measuring commuters' travel times are detailed in Sects 3.1.1 and 3.1.2, respectively.

**3.1.1 Residential and workplace identification.** Accurate identification of residential locations is crucial for precise commuting mobility analysis. As described in Sect 3.1.1, actual places of residence were initially approximated within a defined zone. However, due to potential misalignment between the grid system used by Agoop Corp. and the 1 km grids in this study, the actual home location and recorded home location could differ by up to 1 km. This distance corresponds to a maximum walking time of approximately 15 minutes, based on observed human walking speeds (1.1–1.4 m/s) in prior studies [8,28]. Given that 15 minutes is a non-negligible duration for commuting mobility studies, further refinement of residential locations was necessary to maintain accuracy while respecting user privacy.

To address this issue, instead of applying the home location provided by the data, we implemented a 100-meter grid resolution, providing a finer distinction between home and work locations than the initial 1 km grids. This resolution was chosen as an optimal balance between improved accuracy and data privacy concerns. A user's home grid was identified based on two criteria: 1) the first GPS record at 5 a.m. within the user's home city (chosen as a time when most residents are likely to be at home), and 2) the grid where the user accumulated over 4 hours of stay time within a single day. This approach significantly reduced potential errors in location identification that could impact commuting time calculations.

For workplace identification, the following criteria were used: 1) the user must have accumulated at least 5 hours of stay time within a grid during the day, 2) the grid must be within the identified work city, and 3) the grid must not coincide with the identified home grid. For users who might have more than one workplace or whose workplace is artificially divided by grid boundaries (approximately 3%), we used the first workplace visited after leaving home to capture accurate commuting distance and time (Sect 3.1.2). This assumption is significant for our model as it ensures that each commuter is uniquely assigned to one home and one workplace . This one-to-one mapping at the individual level is fundamental for the internal consistency of the gravity model-based approach, preventing a single commuter from being counted as an outflow to multiple destinations (Sect 3.2).

After processing the data, each geolocation log was assigned a status of home, work or other. Users whose trajectories included both home and work statuses were classified as commuters. As a result, an average of 60% of users in the six selected metropolitan areas were identified as commuters, with their residential and workplace locations successfully marked. This ratio is approximately 10% higher than the commuting rate reported by the Statistics

Bureau of Japan [29]. Considering the known bias in smartphone usage data, this discrepancy is plausible and underscores the reliability of our dataset and methods.

S2 Fig (S1 File) further validates the reliability of this method by mapping the daily trajectory of a typical identified commuter, clearly illustrating the commuting pattern of leaving home in the morning and returning home in the evening.

**3.1.2 Commuting distance and time.**   Based on the processed geolocation records with assigned statuses, each user's commuting time was calculated as the shortest time interval between the home and work logs. To verify the accuracy of these estimates, we plotted the cumulative distribution function (CDF) of the estimated commuting times in S3 Fig (S1 File), which revealed an exponential distribution with a tendency toward shorter commutes. To minimize the impact of outliers and fluctuations in commuting time, we capped the maximum commuting time at 120 minutes (2 hours), capturing over 99.5% of the data. In the Tokyo metropolitan area, the average commuting time was approximately 39.9 minutes, the longest among the target areas. According to the Statistics Bureau of Japan [29], the average commuting time for household heads is 45.9 minutes. This difference can likely be attributed to the inclusion of non-household heads in our dataset, who may have shorter commutes. For instance, younger workers and part-time employees, who are less likely to be household heads, often live in rented accommodations closer to their workplaces. This living arrangement may contribute to the overall shorter average commuting time observed in our study. Additionally, our study capped the commuting time at 120 minutes to minimize the impact of outliers, which could also account for some of the observed differences. This methodological choice ensures that the analysis remains focused on typical commuting behaviors, aligning with the study's objective to capture everyday mobility patterns.

Commuting distance was estimated by calculating the Euclidean distance between the centroid of the identified home grid and the work grid. As shown in S3 Fig (S1 File), the average commuting distance in the Tokyo metropolitan area was approximately 14.7 km, the longest among the studied areas.

## 3.2 The SSUG model

In this study, we propose two key improvements to the traditional gravity model. In the first stage, the distance decay function, which typically relies on commuting distance, is replaced by commuting time to 1 minute's accuracy, as shown in Eq (1). This adjustment reflects the idea that commuting time is a more accurate representation of the deterrence factor affecting commuter behavior, given the high-resolution time data available in this study.

$$f_{ij} = K \frac{h_i^{\alpha} w_j^{\beta}}{t_{ij}^{\delta}}$$

(1)

In Eq (1), $f_{ij}$ represents the commuting flow, $h_i$ denotes the residential population observed in grid $i$, $w_j$ signifies the employee population observed in grid $j$, and $t_{ij}$ indicates the estimated commuting time between grid $i$ and grid $j$. The parameters $\alpha$, $\beta$, and $\delta$ are scaling exponents, while $K$ is a constant value.

Furthermore, we implemented threshold values $h_c$ and $w_c$, as shown in Table 1, to determine whether the residential population ($h_i$) and the employee population ($w_j$) at each grid exceed these thresholds. This classification divides the home (origin) and work (destination) grids into two distinct types: urban and suburban. Consequently, commuting trips are categorized into four types: $f_{ij}$ from $O_S$ to $D_S$, $O_S$ to $D_U$, $O_U$ to $D_S$, and $O_U$ to $D_U$, where $O$ represents origin, $D$ represents destination, $S$ denotes suburban, and $U$ denotes urban.

**Table 1. Types of origin-destinations based on population scales.**

| Origin-Destination types | Population conditions |
|---|---|
| $Origin_{urban}$ ($O_U$) | $h_i \geq h_c$ |
| $Origin_{suburb}$ ($O_S$) | $h_i \leq h_c$ |
| $Destination_{urban}$ ($D_U$) | $w_j \geq w_c$ |
| $Destination_{suburb}$ ($D_S$) | $w_j \leq w_c$ |

Table notes: $h_i$ and $w_j$ represent the population of origin and destination zones, respectively. $h_c$ and $w_c$ are the population thresholds for classifying urban and suburban areas.

As shown in Eq (2), for flows originating from suburban areas ($O_S$), we apply the parameter $\alpha_1$, while for flows from urban areas ($O_U$), we use $\alpha_2$. Similarly, for flows destined for suburban areas ($D_S$), we apply the parameter $\beta_1$, and for urban destinations ($D_U$), we use $\beta_2$. Models 1 through 4, each depict a different combination of population scales at the origin and destination, providing context-sensitive predictions of commuting flows. The variation and improvement of the SSUG model are illustrated in Sects 4.1 and 4.2.

$$\text{Model}_1 : f_{ij} = K_1 \frac{h_i^{\alpha_1} w_j^{\beta_1}}{t_{ij}^{\delta}}, \quad \text{where } h_i \leq h_c; w_j \leq w_c$$

$$\text{Model}_2 : f_{ij} = K_2 \frac{h_i^{\alpha_1} w_j^{\beta_2}}{t_{ij}^{\delta}}, \quad \text{where } h_i \leq h_c; w_j \geq w_c$$

$$\text{Model}_3 : f_{ij} = K_3 \frac{h_i^{\alpha_2} w_j^{\beta_1}}{t_{ij}^{\delta}}, \quad \text{where } h_i \geq h_c; w_j \leq w_c$$

$$\text{Model}_4 : f_{ij} = K_4 \frac{h_i^{\alpha_2} w_j^{\beta_2}}{t_{ij}^{\delta}}, \quad \text{where } h_i \geq h_c; w_j \geq w_c$$

$$(2)$$

To ensure model continuity and consistency, we impose two additional constraints:

$$\text{Constraint 1}: \text{When } h_i = 1 \text{ and } w_j = 1, f_{ij} = 1$$
$$\text{Constraint 2}: \text{When } h_i = h_c \text{ and } w_j = w_c, \text{Model}_1 = \text{Model}_2 = \text{Model}_3 = \text{Model}_4$$

$$(3)$$

Constraint 1 states that when an origin $i$ has one commuter ($h_i = 1$) and a destination $j$ also has one job ($w_j = 1$), the observed flow between them must be one ($f_{ij} = 1$). The consistency of this constraint is guaranteed by our preprocessing rule for handling multiple workplaces, as detailed in Sect 3.1.1. For instance, in the scenario where a single resident from zone $i$ might travel to two potential single-job zones, $j$ and $k$, our methodology resolves the ambiguity as follows:

- A single resident is located in zone $i$ ($h_i = 1$).
- This resident first travels to a workplace in zone $j$ and later to a workplace in zone $k$.
- Per our rule, only the first workplace is considered. The input data for the model is therefore processed as $w_j = 1$ and the contribution to $w_k$ from this resident is 0.
- Consequently, the model correctly enforces $f_{ij} = 1$ and $f_{ik} = 0$, preserving the flow constraint that the total outflow from zone $i$ is 1.

This ensures our model remains internally consistent and that Constraint 1 provides a valid baseline for normalization. It also provides a consistent baseline for the constant $K(K_1)$ across all four model types, ensuring continuity and model interpretability as shown below.

Together, these constraints transform our initial formulations into more complex expressions with consistent behavior at boundary conditions (for clarification, please see Sect 11 (S1 File)).

$$\text{Model}_1 : f_{ij} = K \frac{h_i^{\alpha_1} w_j^{\beta_1}}{t_{ij}^{\delta}}, \quad \text{where } h_i \leq h_c; w_j \leq w_c$$

$$\text{Model}_2 : f_{ij} = K \cdot \frac{w_c^{\beta_1}}{w_c^{\beta_2}} \cdot \frac{h_i^{\alpha_1} w_j^{\beta_2}}{t_{ij}^{\delta}}, \quad \text{where } h_i \leq h_c; w_j \geq w_c$$

$$\text{Model}_3 : f_{ij} = K \cdot \frac{h_c^{\alpha_1}}{h_c^{\alpha_2}} \cdot \frac{h_i^{\alpha_2} w_j^{\beta_1}}{t_{ij}^{\delta}}, \quad \text{where } h_i \geq h_c; w_j \leq w_c$$

$$\text{Model}_4 : f_{ij} = K \cdot \frac{h_c^{\alpha_1}}{h_c^{\alpha_2}} \cdot \frac{w_c^{\beta_1}}{w_c^{\beta_2}} \cdot \frac{h_i^{\alpha_2} w_j^{\beta_2}}{t_{ij}^{\delta}}, \quad \text{where } h_i \geq h_c; w_j \geq w_c$$

(4)

We initially formulated our models with distinct parameters $K_1$-$K_4$ (Eq (2)). However, to maintain prediction continuity across model boundaries and ensure consistent behavior, we introduced two constraints (Eq (3)). When these constraints are applied, the models transform into their final form (Eq (4)) with a single constant $K$ and additional scaling factors that guarantee smooth transitions between urban and suburban zones. Consequently, the SSUG model comprises eight parameters: $(h_c, w_c, \alpha_1, \alpha_2, \beta_1, \beta_2, \delta, K)$, and this multidimensional parameter estimation problem is solved by Bayesian optimization (BO) (Sect B.2).

Also, since each grid can be regarded as both a residential area and a workplace, the grid itself can be classified into four types: $h_s w_s$, $h_s w_u$, $h_u w_s$, and $h_u w_u$. The classification results are presented in Sects 4.4 and 5.

The SSUG model introduces two main points of novelty. First, rather than dividing a region based on predetermined qualitative standards of urban structure, the SSUG model offers a quantitative approach by integrating segregation parameters directly into the calibration process. This approach not only presents a new application of the gravity model, where the calibration of its parameters can be used to detect segregation mathematically, but also provides a practical, quantitative method for measuring the urbanity of a metropolitan area. For example, the Densely Inhabited District (DID) in Japan is officially defined by a population density of 4,000 people per 1 km$^2$, along with the population of adjacent regions [21]. The SSUG model offers a quantitative reference that aligns with this official definition, thereby enhancing its applicability in urban studies.

Second, the calibrated parameters above and below the thresholds $h_c$ and $w_c$ have meaningful interpretations. The parameters of gravity models, particularly the scaling exponents represented by $\alpha$ and $\beta$, have long been discussed in the literature for their role in indicating the strength of attraction between locations [30]. The SSUG model allows for a direct comparison of these parameters when applied to different types of commuting trajectories, providing valuable insights into how commuting patterns vary across different origin and destination population scales. Notably, this approach emphasizes the significant flow contribution in highly populated areas. We also discussed the CES form of the gravity model [31] in Sect 10 (S1 File).

**3.2.1 OD and the commuting time per OD.** In the gravity model, Origin-Destination (OD) pairs are used as the fundamental units of human mobility. Here, $f_{ij}$ represents the commuting volume between a specific origin mesh $i$ and destination mesh $j$.

When the model uses distance as the cost-effect factor, it is typically calculated as the distance between the centroid of the origin mesh and the destination mesh, with each OD corresponding to a single distance value. However, commuting time is dependent on the behavior of individual commuters, meaning that each OD pair can contain multiple commuting times.

However, commuting time is dependent on the behavior of individual commuters, meaning that each OD pair can contain multiple commuting times. This real-world variability stems from several factors:

- Transportation Mode: For a single OD pair, individuals may use various modes with different time profiles, such as private cars, motorcycles, bicycles, or different public transit options (e.g., local vs. express trains).
- Route and Service Choice: Even within a single mode, choices matter. Drivers may choose between highways and local roads, while transit users may select different lines or transfer points.
- Real-time Conditions: Road traffic congestion is a major source of variability for drivers. Similarly, public transit is subject to delays, crowding, and service schedules that vary by time of day.
- Individual Behavior: Raw trajectory data can capture non-commuting detours, such as stopping for errands, which increase total travel time.

Despite this variability, we assume that the fundamental deterrence for an OD pair is best represented by the shortest possible travel time. In this study, we assume that commuters consistently choose the shortest path between an origin and a destination, following the principle that no commuting-unrelated behaviors are undertaken during the trip. This assumption aligns with findings from previous research, which show that commuters tend to optimize their travel time [32]. Moreover, the observed exponential distribution of commuting times (S3 Fig in S1 File) supports this assumption by suggesting a preference for shorter travel times among commuters. This approach also helps avoid overestimating commuting time due to traffic congestion [33]. Based on this, we assume that the commuting time for an OD pair corresponds to the shortest travel time among all the trips observed for that pair, as expressed in Eq (5):

$$t_{ij} = \min\{t_{ij}^k \mid k \in \{1, 2, \ldots, n_{ij}\}\} \tag{5}$$

where $t_{ij}^k$ represents the travel time of the $k$-th trip between origin mesh $i$ and destination mesh $j$, and $n_{ij}$ is the total number of trips between these two meshes. Robustness tests for this assumption, where minimum values are replaced with mean and median values, are presented in the Supplement.

**3.2.2 Multidimensional parameter estimation method.** The eight parameters $(h_c, w_c, \alpha_1, \alpha_2, \beta_1, \beta_2, \delta, K)$ were estimated using Bayesian optimization (BO) [34]. Bayesian optimization is a sequential model-based optimization method that is particularly effective for optimizing expensive functions with boundaries and conditions. For each step of BO, it builds a probabilistic model of the objective function, a Gaussian process (GP) is trained using all previous observations of the model function, and uses this model to select the most promising parameters for evaluation. The detailed steps are written in Appendix B.2.

For this optimization problem in the SSUG model with eight dimensions, the objective function is set as below in Eq (6):

$$\text{Objective function: } \frac{\sum_{i,j} \log\left(\frac{\hat{f}_{ij}}{f_{ij}}\right)}{\text{No. of ODs}} \tag{6}$$

Where $f_{ij}$ denotes the empirical commuting volume between origin mesh $i$ and destination mesh $j$, while $\hat{f}_{ij}$ is the estimated commuting flow. To obtain reliable results, we ran the optimization for each metropolitan area 1,000 times, and adopting the results with the lowest objective function value.

### 3.3 Model validation check

To determine the appropriateness of dividing the gravity model into two distinct segments within a metropolitan area, we employed the F-test. Commonly used in regression analysis and analysis of variance (ANOVA) [35], the F-test was applied to assess whether the variance in residuals significantly decreases when the dataset is split into two segments, compared to using single model for the entire dataset. This approach is consistent with methodologies used in previous research, which evaluate whether two parts of a sample dataset exhibit different correlations with a particular variable [36]. Detailed formulas and calculations are provided in Appendix A.

In the context of the gravity model, which exhibits a linear relationship in log-log scale, as shown in Eq (7), the F-test is particularly suitable for assessing the linear relationships between the variables $f_{ij}$ and $h_i$ (where the slope indicates $\alpha$), as well as $f_{ij}$ and $w_j$ (where the slope indicates $\beta$). This statistical test examines whether, after dividing the variables $h_i$ and $w_j$ into two segments (separated by thresholds $h_c$ and $w_c$), the variance in their linear relationships with $f_{ij}$ significantly differs from the variance observed in a single, undivided model.

$$\log(f_{ij}) = \log(K_1) + \alpha \log(h_i) + \beta \log(w_j) - \delta \log(t_{ij}) \tag{7}$$

The test thus indicates whether the division of the dataset leads to an improvement in model fit.

The parameters of the linear models were estimated using the Least Squares Method. For each possible combination of $h_c$ and $w_c$ values, we conducted an F-test and calculated the corresponding F-values. By comparing these F-values with the critical values at a 0.05 confidence level, we statistically determined whether dividing the population scales into two segments provided a better fit for the data. Specifically, we employed F-tests to statistically validate our core hypothesis that structural breaks exist at the thresholds $h_c$ and $w_c$. The F-test determines whether the relationship between population variables ($h_i$ or $w_j$) and commuting flows ($f_{ij}$) differs significantly for values below versus above each threshold. A significant F-statistic indicates that the data should indeed be treated as two distinct regimes, thereby justifying our segmented modeling approach based on population scale thresholds.

It is important to note that these threshold validation tests serve solely to confirm the existence of structural breaks in the data—the parameters derived from these F-tests were not incorporated into the final SSUG model fitting or evaluation procedures described below.

### 3.4 Measuring the similarity between the estimated flows and empirical data

To compare the performance of the SSUG model, the commuting time-based gravity model, and empirical data, the evaluation and validation of the model's improvement are expressed through the Common Part of Commuters (CPC) index [37]. The CPC index is a modified version of the Sørensen Similarity Index (SSI), widely used in human mobility-related literature to quantify the similarity between predicted human flows and observed human flow data [8]. The CPC is calculated as shown in Eq (8):

$$\text{CPC} = \frac{2 \sum_{i=0}^{N} \sum_{j=0}^{N} \min\left(f_{ij}, \hat{f}_{ij}\right)}{\sum_{i=0}^{N} \sum_{j=0}^{N} \left(f_{ij} + \hat{f}_{ij}\right)} \tag{8}$$

Where $f_{ij}$ denotes the empirical commuting volume between origin mesh $i$ and destination mesh $j$, while $\hat{f}_{ij}$ is the estimated commuting flow. $N$ is the number of OD pairs in one metropolitan region. The value of CPC ranges from 1, when each estimated $\hat{f}_{ij}$ equals $f_{ij}$, to 0, when the estimation scenario is the opposite.

## 4 Results

### 4.1 Statistical validation of the SSUG model

Our study identified significant deviations in the traditional gravity model's performance when applied to both sparsely and densely populated urban areas (see S4 Fig (S1 File)), which aligns with previous research findings [15]. Specifically, the model tends to overestimate commuting flows in sparse areas while underestimating them in denser regions [13]. These discrepancies highlight the need for a more refined approach that can address the varying characteristics across different regions of the metropolitan area [15].

This issue is rooted in the gravity model's format, Eq (1), as it is built on the flow $f_{ij}$'s relationship with $h_i$, $w_j$, and $t_{ij}$. The deviations associated with population scale suggest that the relationships between $f_{ij}$ and $h_i$, as well as $f_{ij}$ and $w_j$, are inconsistent. In Fig 1(a) and 1(b), we illustrate these relationships by plotting the data for Tokyo's city center (the 23 special wards of central Tokyo) separately from other regions. The human flow volume $f_{ij}$ in the central part shows significantly higher median values and outliers compared to other regions, which explains why the gravity model's performance is scale-dependent. When applying ridge regression or linear regression on the log-log format of the gravity model (7), these outliers are often treated as residuals. However, these extreme flows are vital for human mobility analysis and significantly impact model predictions.

To address these issues, we hypothesized that segregating residential and office areas into urban and suburban categories based on residential threshold $h_c$ and employee threshold $w_c$ would improve the model's accuracy. To verify this hypothesis, we tested whether two distinct datasets, based on $h_c$ and $w_c$, have significantly different correlations with the gravity model's parameters $\alpha$ and $\beta$. This analysis aimed to determine if the exponents for the origin and destination sides exhibit distinct values when appropriate division thresholds are applied.

The F-test was employed to assess whether dividing the dataset based on the thresholds $h_c$ and $w_c$ leads to different correlations with the dependent variable $f_{ij}$, as detailed in the methodology Sect 3.3. A significant F-value beyond the critical value indicates substantial improvement regarding the residuals of the dataset when regressed with distinct slopes versus a single slope. This suggests that using distinct slope regressions explains meaningful variance beyond random chance. We used the F-test to determine whether population segmentation by

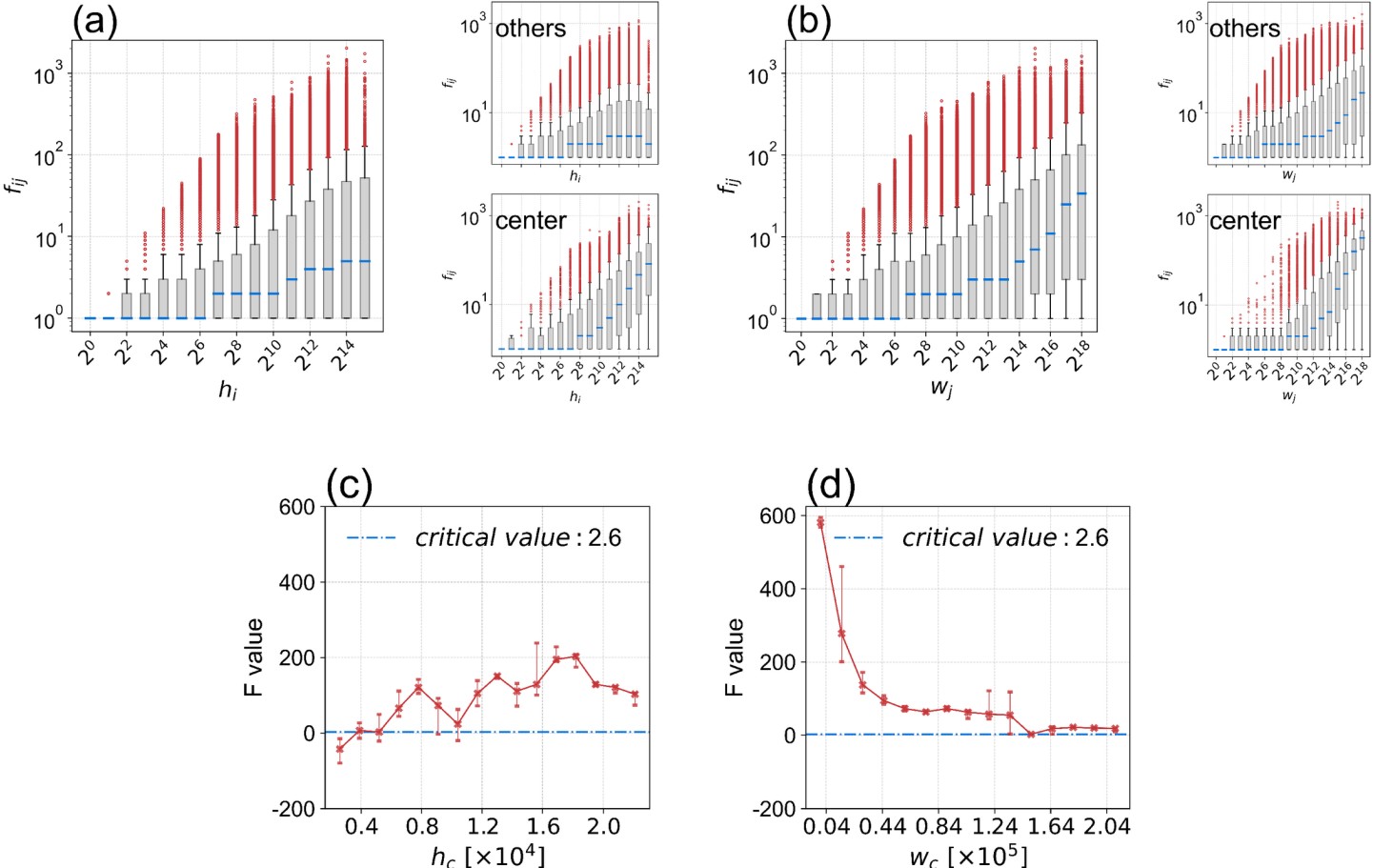

**Fig 1. Flow distribution patterns and F-value analysis across population scales.** (a) and (b) display the box-and-whisker plots of $f_{ij}$ across different scales of $h_i$ (a) and $w_j$ (b), with separate subplots for the central Tokyo area (23 special wards, see S1 Fig (S1 File)) and other regions shown in the right sides. The box in each plot represents the interquartile range (IQR), spanning from the first quartile (Q1, 25th percentile) to the third quartile (Q3, 75th percentile), while the blue line inside the box marks the median (50th percentile). The whiskers extend up to 1.5 times the IQR, and any points beyond are considered outliers. In the central Tokyo region, there is a notable concentration of large flows (larger median values and outliers), especially compared to other regions, indicating distinct $f_{ij}$ distributions based on residential (a) and employee population scales (b). (c) and (d) show the F-values across varying threshold values of $h_c$ and $w_c$, respectively. The x-axes represent segregation points for $h_c$ (c) and $w_c$ (d) along the population scale, while the y-axes display the corresponding F-values.

$h_c$ and $w_c$ significantly enhances the model by allowing distinct values for $\alpha$ and $\beta$ in each segment. Since the specific values of $h_c$ and $w_c$ were unknown, we systematically tested a range of possible thresholds, from minimum to maximum, resulting in two distinct sample datasets for each threshold. The results of this analysis are shown in Fig 1(c) and 1(d).

In summary, the larger the F-value on the y-axis, the smaller the residual values in the two-part linear relationships between $f_{ij}$ and $h_i$ (Fig 1(a)), as well as $f_{ij}$ and $w_j$ (Fig 1(b)), compared to the single, unsegmented forms. The analysis shows that for both residential and workplace settings, setting thresholds $h_c$ and $w_c$ results in a significant decrease in residual values, indicating two distinct segments within the dataset for both $h_i$ and $w_j$ in relation to their correlations with $f_{ij}$.

Thus, the F-test was chosen for its ability to directly compare single-slope and double-slope regressions, ensuring the dataset's segmentation is statistically justified. While other methods like AIC, BIC, Likelihood Ratio Test, and Cross-Validation could have been considered, the

F-test was selected for its straightforward application within the linear regression framework of the gravity model. This approach led to more accurate estimates of $\alpha$ and $\beta$, validating that segregating residential and office areas into distinct categories enhances model calibration with reduced residuals. Consequently, the SSUG model provides a more precise and reliable analysis.

## 4.2 Predicting human mobility flows based on the SSUG model

Based on the results of Sect 4.1, it is statistically valid to treat the residents' and employees' population datasets as two distinct segments, separated by their respective scales. We introduced thresholds $h_c$ and $w_c$ to optimize this classification, with the results summarized in Table 2, and the results' robustness analysis is presented in Sect 8 (S1 File). Grids with residential populations exceeding $h_c$ are classified as urban residential areas, while those below this threshold are designated as suburban residential areas. Similarly, grids where the employee population surpasses $w_c$ are categorized as urban workplace areas, with those below this threshold classified as suburban workplace areas.

In the case of the Tokyo metropolitan area, the analysis revealed that urban residential grids account for only 1.16% of the total grid area, while urban workplace grids make up 1.92%. This distribution underscores the highly concentrated nature of urban core areas compared to suburban regions.

Furthermore, the commuting flows were estimated using the SSUG model for the study period. For comparison, the performance of the previous study's distance-based gravity model Eq (9) [5,15,16] and the commuting-time-based gravity model (Eq (1)) under the same conditions was also assessed.

$$f_{ij} = K_1 \frac{h_i^\alpha w_j^\beta}{d_{ij}^\gamma} \tag{9}$$

In this study, we adopted a power-law form of the Euclidean distance $d_{ij}$ as the distance decay function. The commuting flows estimated by the models were denoted as follows: $f^s$ for the SSUG model, $f^c$ for the commuting-time-based gravity model, and $f^d$ for the Euclidean distance-based gravity model. The empirically observed commuting flows extracted from the data are represented by $f$.

The performance of these models ($f^s, f^c, f^d$) was compared, and the results are illustrated in Figs 2 and 3. The accuracy of the models' predictions was evaluated using the Common Part of Commuters (CPC) index (see Sect 3.4), with the results summarized in Table 2. The SSUG model achieved the highest CPC value of 0.514, demonstrating superior accuracy

**Table 2. Comparison of model parameters and performance.**

| Model | $h_c$ | $w_c$ | $\alpha$ | $\beta$ | $\delta$ | CPC |
|---|---|---|---|---|---|---|
| Gravity model-Distance | - | - | $\alpha_d = 0.140$ | $\beta_d = 0.300$ | $\delta_d = 0.387$ | 0.27 |
| Gravity model-Commuting time | - | - | $\alpha_c = 0.134$ | $\beta_c = 0.286$ | $\delta_c = 0.669$ | 0.37 |
| SSUG model | 12143 | 11445 | $\alpha_1 = 0.009$ | $\beta_1 = 0.010$ | $\delta_s = 0.981$ | 0.41 |
| | (98.83%) | (98.07%) | $\alpha_2 = 1.912$ | $\beta_2 = 1.312$ | | |

Table notes: The results showed that the CPC of the SSUG model (CPC = 0.41) was significantly larger than that of un-segregated distance-based gravity model (CPC = 0.27) and the commuting time-based gravity model (CPC = 0.37).

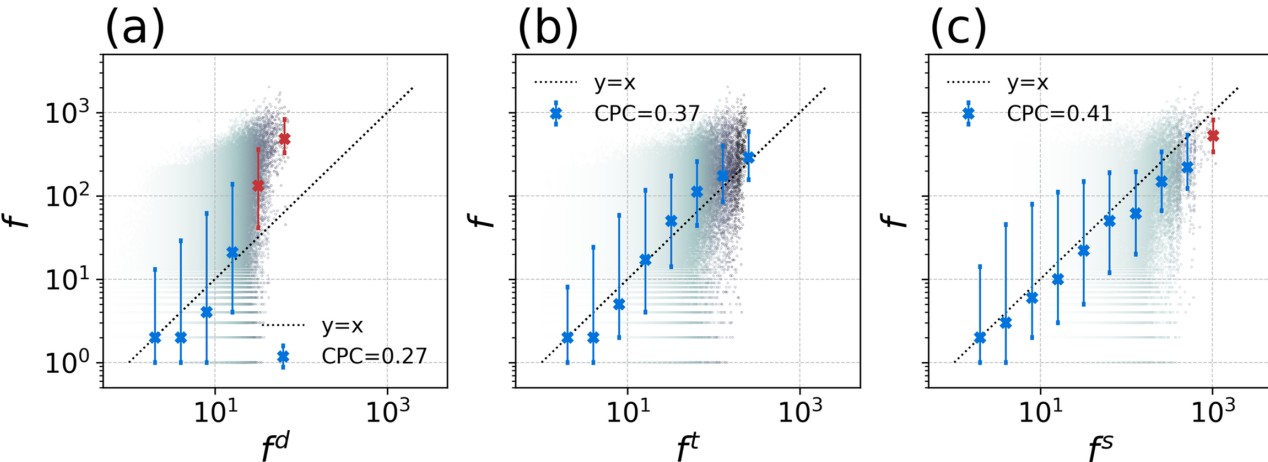

**Fig 2. Comparison between empirical and model-predicted commuting flows.** This figure compares the empirical commuting flows (f) observed from the data with the predicted commuting flows from three different models: (a) Euclidean distance-based gravity model ($f^d$), (b) commuting-time-based gravity model ($f^t$), and (c) SSUG model ($f^s$). Each scatter point represents a pair of predicted and actual commuting flows, with the points divided into bins based on the predicted values. The markers indicate the median real flow in each bin, while the error bars show the range between the 1st and 3rd quantiles within each bin. The black dotted line represents the line where predicted flows equal the real flows (f), serving as a reference for accuracy. Underestimation occurs when the markers are above the dotted line, while overestimation occurs when they are below it. Error bars are colored red when there is significant misestimation, where the y-values lie outside the 1st and 3rd quantiles of the x-values' bins, and blue otherwise. This comparison highlights the performance differences between the models, with the SSUG model (c) showing the best alignment with the empirical data, indicating fewer instances of significant overestimation or underestimation compared to the other models.

compared to the other models. In the Tokyo metropolitan area, the Euclidean distance-based gravity model yielded the lowest CPC value (0.266), while the commuting-time-based model improved predictions with a CPC of 0.371. Although commuting time is generally a more logical factor than distance for capturing decay in commuting flows, the SSUG model still outperformed both alternative models in terms of predictive accuracy.

To further assess the models' fitness across different scales of $f_{ij}$, Fig 2 presents the results using error bars, with bins set based on commuting volumes. The CPC values from Table 2 are also indicated in the top left corner. In Fig 2(a) and 2(b), representing the Euclidean distance-based and commuting-time-based gravity models, larger commuting flows tend to be under-estimated. In contrast, the SSUG model's predictions (Fig 2(c)) demonstrate superior performance, particularly in compensating for inaccuracies in larger commuting flows, which are better captured by the SSUG model compared to the commuting-time-based model.

The Euclidean distance-based model (Fig 2(a)) exhibits a clear tendency to overestimate flows in sparsely populated areas and underestimate them in denser regions. While adjusting the distance-decay factor to commuting time (Fig 2(b)) improves accuracy, it still struggles with underestimation for large flows and overestimation for sparse flows. In contrast, the SSUG model (Fig 2(c)) provides a more balanced prediction, with only slight overestimations for extremely large flows and more accurate predictions for sparse flows and overall commuting patterns.

These results indicate that the SSUG model's incorporation of spatial segmentation significantly enhances its predictive accuracy. Although minor overestimations for large flows persist, the SSUG model effectively reduces both overestimation and underestimation errors across varying commuting volumes. This analysis yields two important insights: first, in

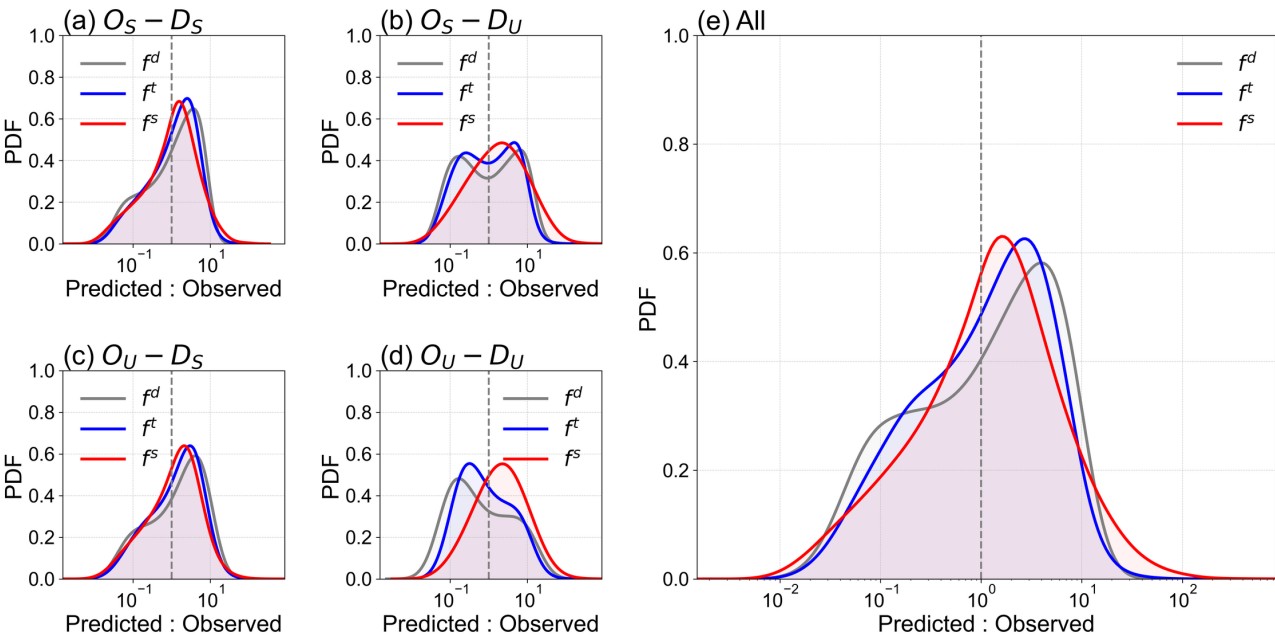

**Fig 3. Probability density functions of model performance across commuting trip types.** This figure displays the probability density function (PDF) of the ratio between predicted and observed trip counts across different commuting trip types within the Tokyo metropolitan area. The comparison is made between three models: the distance-based gravity model $f^d$ (grey), the commuting-time-based gravity model $f^c$ (blue), and the SSUG model $f^s$ (red). (a)–(d) depict the PDF for specific trip types, while (e) provides an overall distribution for all trip types combined. The SSUG model consistently shows peaks closer to 1 across all trip types, indicating a higher accuracy in predictions compared to the other models. For suburban destination trips ((a) and (c)), the SSUG model minimizes the common overestimation seen in the traditional models by more accurately capturing the appeal of suburban areas. Conversely, the $O_U$–$D_U$ trip type in (d) highlights the SSUG model's ability to reduce underestimation errors, which are more prevalent in dense urban areas.

metropolitan areas, commuting time exerts a stronger influence on the reluctance to undertake longer trips; second, the volume of commuting flows is differentially affected by the population scales of the origin and destination.

As shown in Fig 2, the overall model fit primarily demonstrates how well each model performs in a general case. However, it is essential to evaluate the SSUG model's performance across different trip types in comparison to other models. To assess this, we applied the ratio of predicted to observed commuting flow as an indicator to understand the distribution of overestimation (ratio >1) and underestimation (ratio <1) for each trip type. Fig 3a–3d present the probability density function (PDF) of this ratio for various trip types, while Fig 3e offers an aggregated view of the ratio's distribution.

The SSUG model (red lines) consistently demonstrates a higher proportion of accurate predictions across all four trip types, with the peak of the PDF closest to 1, indicating where the predicted value equals the observed value. In contrast, while the commuting-time-based gravity model outperforms the distance-based gravity model, both tend to overestimate commuting flows within suburban areas ($O_S D_S$, Fig 3(a)) and underestimate flows within urban areas ($O_U D_U$, Fig 3(d)). This discrepancy is likely due to the uniform scaling exponents $\alpha$ and $\beta$, which do not account for the varying influence of origin and destination characteristics.

For trip type $O_U D_S$ (Fig 3(c)), while the SSUG model performs best, all three models show a tendency to overestimate flows. This may suggest the presence of additional decay factors beyond those captured by the gravity model, such as limited job positions or infrastructure constraints in suburban workplace areas.

In the case of trip type $O_S D_U$, where flows originate from less populated residential areas and terminate in more densely populated office regions, both unsegregated models ($f_d$ and $f_c$) display instances of both overestimation and underestimation. The SSUG model, however, provides the most significant improvement. Overestimation of $f_d$ and $f_c$ may be explained by the estimation method used (see Sect 3.2.1), which assumes that all commuters sharing the same origin and destination (OD) take the shortest commuting time. In $O_S$ regions, a limited number of commuters might have access to private transportation, such as cars or bicycles, which enables shorter commute times, while the majority rely on public transportation, resulting in longer commute times.

Underestimation, on the other hand, may arise from limited transportation resources in suburban regions, which tend to concentrate job locations and restrict residents to specific work destinations. This could result in actual flow volumes that are higher than those calculated by the model-due to the aggregation of commuting patterns around a few key destinations.

In conclusion, the SSUG model outperforms the traditional, unsegregated gravity models by delivering more accurate and balanced predictions across different trip types. The incorporation of commuting time proves to be a more effective decay factor for flows in metropolitan areas. The SSUG model's ability to reduce both overestimation and underestimation errors, particularly in suburban and urban commuting scenarios, highlights its effectiveness in capturing complex urban mobility patterns.

## 4.3 Spatial distribution of commuting propensities reflected by the SSUG model

In this study, we enhanced the gravity model by incorporating threshold parameters $h_c$ and $w_c$ during the calibration process. We utilized the gravity model not only to measure commuting patterns but also as a tool to classify urban and suburban areas. Moreover, we found that the power-law exponents, representing heterogeneity, offer meaningful insights into commuting behaviors within metropolitan regions.

Our primary focus was on the real-world implications of the exponents $\alpha$ and $\beta$, which represent the attractive forces between origins and destinations in the gravity model. As prior research [38] suggests, larger values of $\alpha$ reflect a stronger propensity for residents to commute to a limited set of preferred workplaces, while larger values of $\beta$ indicate that employees are more likely to commute from a limited number of preferred residential areas. According to our calibration results (Table 2); full regional results in S1 Table (S1 File)), we observe that $\alpha_2 > \alpha_1$ and $\beta_2 > \beta_1$ across all six selected Japanese metropolitan areas. This implies that urban areas (denoted by $\alpha_2$ and $\beta_2$) exhibit stronger commuting propensities than suburban areas. In practical terms, this means that urban residential areas tend to have more concentrated commuting destinations (as seen from $\alpha_2 > \alpha_1$), and urban workplaces draw commuters from more concentrated origins (as indicated by $\beta_2 > \beta_1$).

Fig 4 visualizes these findings by mapping the geographical distribution of populations and displaying population density distributions (PDFs) along both longitude and latitude. The plots differentiate between commuters from suburban residential areas ($h_i < h_c$, blue grids in Fig 4a) and urban residential areas ($h_i > h_c$, red grids in Fig 4a), showing their workplace locations in Fig 4c and 4d, respectively. Similarly, the residential origins of commuters working in suburban workplaces ($w_j < w_c$, blue grids in Fig 4b) and urban workplaces ($w_j > w_c$, red grids in Fig 4b) are illustrated in Fig 4e and 4f, respectively.

In Fig 4(a) and 4(b), the geographical layout indicates that urban residential areas and workplaces are predominantly located near the Tokyo CBD (Central Business District)).

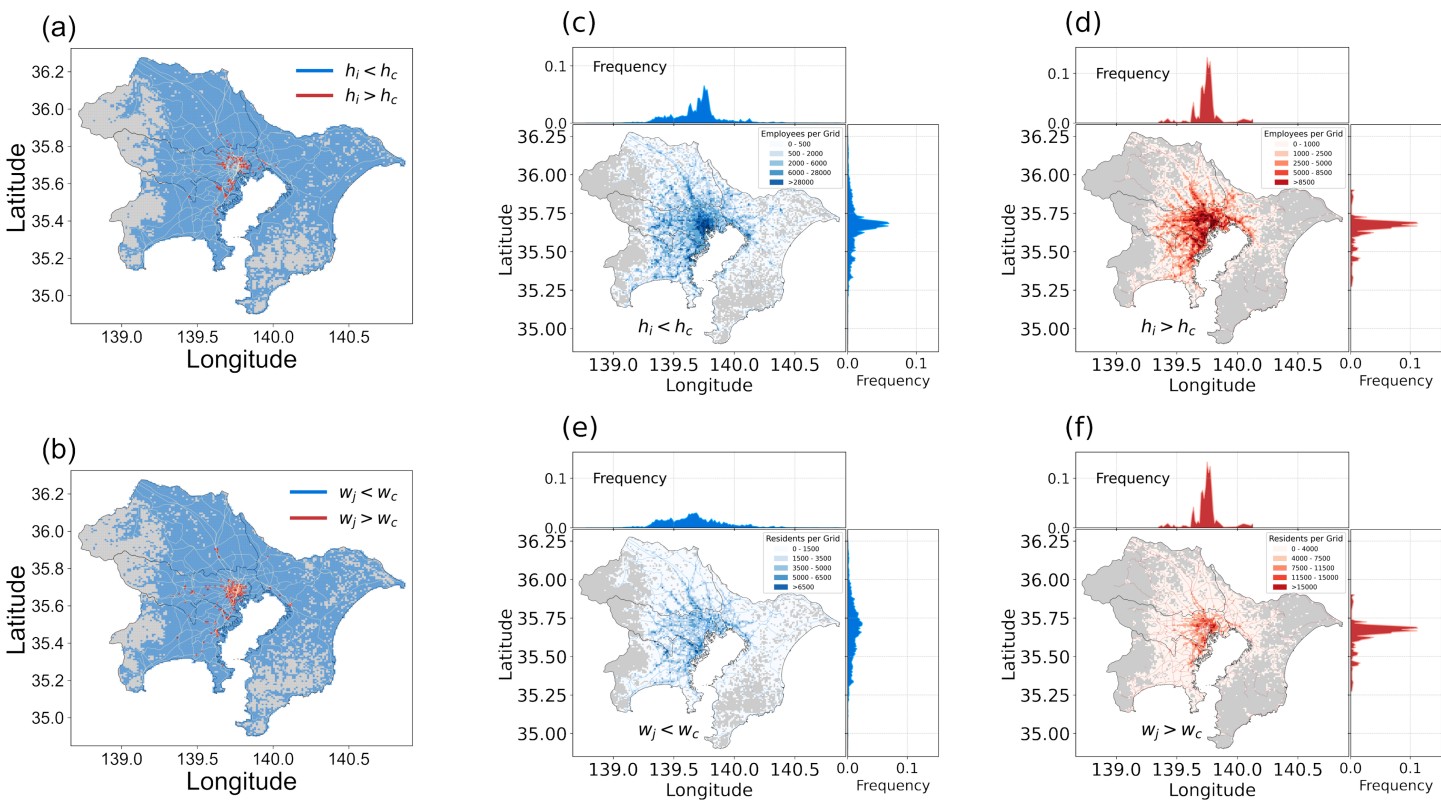

**Fig 4. Spatial distribution of residential and employment populations in Tokyo.** (a) and (b): Maps depicting the spatial distribution of residential (a) and employment (b) populations in the Tokyo metropolitan area in 1 km² grid cells. In (a), blue grids represent areas where residential population $h_i < h_c$, while red grids indicate areas where $h_i > h_c$, showing higher-density residential zones. In (b), blue grids represent areas with employment populations $w_j < w_c$, and red grids highlight areas where $w_j > w_c$, indicating higher-density employment zones. Both maps show central Tokyo concentration for higher-density areas, with lower-density regions on the outskirts. (c)–(f): Distribution of residents' workplaces and employees' residential areas. Suburban areas ($h_i < h_c$ and $w_j < w_c$) are represented in blue, while urban areas ($h_i > h_c$ and $w_j > w_c$) are shown in red. The top and right sides of each subplot show the Probability Density Function (PDF) of population density along longitude and latitude. (c) and (d): Workplace distribution for suburban (c) and urban (d) residents. Suburban residents' workplaces are more evenly dispersed, while urban residents' workplaces are clustered near central commercial hubs. (e) and (f): Residential distribution of suburban (e) and urban (f) employees. Suburban employees' residential areas are widely distributed, while urban employees are densely clustered in central regions. Base map reprinted from [39] under a CC BY license, with permission from Ministry of Land, Infrastructure, Transport and Tourism, original copyright 2008.

Fig 4(d) and 4(f) clearly show that residents and employees in urban areas ($h_i > h_c$ and $w_j > w_c$) have more centrally concentrated destinations and origins. This centralization is evident in the PDFs of population distribution along both longitude and latitude, where urban areas exhibit sharper and more concentrated peaks compared to suburban areas (Fig 4(c) and 4(e)), reflecting the denser population clusters in urban regions.

In summary, the findings reveal that urban origins and destinations exhibit a stronger mutual attraction, as indicated by the greater values of $\alpha_2$ and $\beta_2$ compared to their suburban counterparts, which display more evenly distributed commuting patterns. The mutual attraction in urban areas, as captured by the SSUG model's exponents, suggests that urban residential areas have concentrated commuting destinations, while urban workplaces attract a concentrated set of residents. These results align with previous studies that describe Tokyo's CBD and its surrounding areas as a single functional urban entity composed of the core and its hinterlands [20,40]. Furthermore, the SSUG model serves as a practical tool for identifying city segregation and offers a quantitative approach to assessing urbanization through metrics

like the Densely Inhabited District (DID) index [21], which is crucial for urban planning and policy development.

## 4.4 Region analysis

This study examines six major metropolitan areas in Japan—Tokyo, Osaka, Nagoya, Fukuoka, Sendai, and Sapporo—each differing significantly in population size, area, economic output, and administrative density (see Table 3). These differences provide a robust platform for analyzing diverse commuting patterns. Larger areas like Tokyo and Osaka, with dense urban cores and high economic activity, exhibit more complex structures. In contrast, smaller regions like Sendai and Sapporo show more localized commuting patterns due to their lower density and smaller economic hubs. These variations significantly impact the gravity model's ability to accurately predict commuting flows across different urban contexts.

The complexity of urban structures is illustrated in Fig 5, which presents four grid types (see Sect 3.2): $h_s w_s$, $h_s w_u$, $h_u w_s$, and $h_u w_u$ in the Tokyo metropolitan area. The SSUG model identifies multiple urban office areas ($h_s w_u$ in yellow, $h_u w_u$ in red), surrounded by urban residential areas ($h_u w_s$ in light cyan) and suburban areas ($h_s w_s$ in blue).

Each classification corresponds to a unique urban function with significant real-world implications for planning and mobility. The urban core ($h_u w_u$) represents the primary Central Business Districts (CBDs), such as Chiyoda ward and Shinjuku ward, characterized by a high-density mix of both corporate headquarters and residential populations; the key planning implication here is the need to manage hyper-density with high-capacity, multi-modal public transport. In contrast, areas classified as $h_u w_s$ are dense residential districts with low local employment. A clear example is Kōtō Ward, known for its large-scale apartment complexes. These zones are the epicenters of commuter demand, exporting a massive workforce daily and thus placing immense pressure on public transportation corridors.

These residential areas are complemented by employment magnets ($h_s w_u$), which are secondary urban centers or edge cities like Tachikawa in western Tokyo. These hubs are crucial for decentralization, offering major employment opportunities closer to suburban homes, which can reduce overall commute distances and relieve pressure on the urban core. Finally, the $h_s w_s$ category covers the sparsely populated periphery, such as the mountainous Okutama region. Characterized by low residential and employment densities, planning for these areas must address potential car dependency and the provision of essential local services. This spatial segregation of functions creates significant commuting patterns, as S11 and S12 Figs (S1 File) demonstrate that mismatches between residential and employment populations are the main drivers of commuting flows.

Each distinct function's geographical distribution reveals a core-periphery structure with central business districts, surrounding residential zones, and secondary commercial centers. This pattern aligns with established urban planning theory [41,42]. The identification

**Table 3. Basic information of six Japanese metropolitan areas.**

| Metropolitan area | Area (km$^2$) | Permanent Residents (million) | GDP (JPY trillion) |
|---|---|---|---|
| Tokyo | 13,564 | 36.9 | 193.5 |
| Osaka | 23,334 | 19.1 | 82.3 |
| Nagoya | 21,570 | 11.3 | 57.1 |
| Fukuoka | 4,987 | 5.1 | 19.5 |
| Sendai | 7,282 | 2.3 | 9.6 |
| Sapporo | 3,036 | 2.4 | 8.7 |

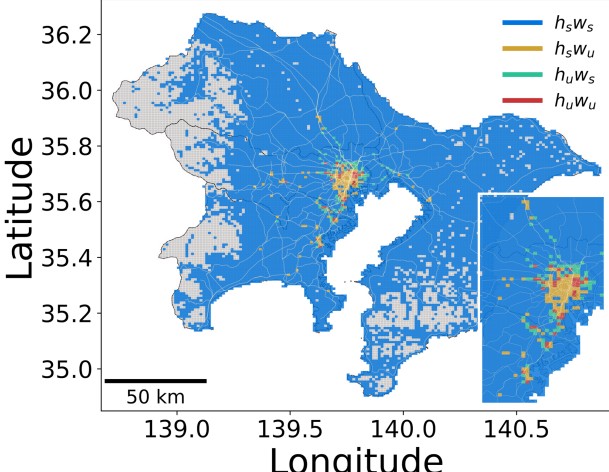

**Fig 5. Spatial distribution of the four mesh types within the Tokyo metropolitan area.** The color coding represents the following categories: blue ($h_s w_s$) for suburban residential areas with suburban workplaces, yellow ($h_s w_u$) for suburban residential areas with urban workplaces, green ($h_u w_s$) for urban residential areas with suburban workplaces, and red ($h_u w_u$) for urban residential areas with urban workplaces. The blue areas dominate the periphery, representing extensive suburban zones, while the red areas, concentrated in the center, indicate dense urban cores where both residents and employees are located. Base map reprinted from [39] under a CC BY license, with permission from Ministry of Land, Infrastructure, Transport and Tourism, original copyright 2008.

of these distinct functional zones is a robust finding; as shown in S8 Fig (S1 File), smaller cities like Fukuoka, Sapporo, and Sendai exhibit similar, albeit more isolated, core-periphery structures, highlighting universal patterns of urban density and connectivity across different metropolitan scales.

Table 4 compares the CPC index across the six metropolitan areas using three models: distance-based, time-based, and SSUG. S6 Fig (S1 File) further supports these results, displaying the PDF of the ratio between predicted and observed trips across regions and trip types. Overall, the SSUG model consistently outperforms traditional models, providing higher CPC values across all regions and trip types. Particularly for urban-to-urban ($O_U – D_U$) trips, the SSUG model shows the most significant improvements, correcting the underestimation seen in other models. For instance, in Sapporo, the SSUG model achieves a CPC of 0.72, compared to 0.16 for the unsegregated commuting time-based model. Sapporo also shows the highest overall improvement (CPC 0.44), especially for urban flows, indicating a strong functional urban core.

However, Sendai exhibited the lowest CPC value (0.34). Though transportation infrastructure and economic diversification affect the situation, it can likely be attributed to its lack of $h_u w_u$ regions, which differs from other selected metropolitan areas, as highlighted in the S8 Fig (S1 File). High-density urban residential and workplace regions ($h_u w_u$) are critical attractors for commuting flows in metropolitan areas [43]. The absence of these regions in Sendai indicates a weaker pull from the central business district, leading to lower predictive accuracy in the SSUG model. As evidenced by the PDF of the ratio between predicted and observed trip counts (see S6 Fig (S1 File)), the SSUG model's prediction for trip type $O_U D_U$ in Sendai is less accurate compared to other regions with better-defined $h_u w_u$ regions, which exhibit stronger urban attraction and better model performance.

**Table 4. CPC values for each metropolitan area by trip type and model.**

| Metropolitan Area | Model | Os - Ds | Os - Du | Ou - Os | Ou - Ou | All |
|---|---|---|---|---|---|---|
| Tokyo | Distance | 0.27 | 0.26 | 0.26 | 0.15 | 0.27 |
| | Time | 0.37 | 0.42 | 0.41 | 0.40 | 0.37 |
| | SSUG | **0.40** | **0.45** | **0.50** | **0.59** | **0.41** |
| Osaka | Distance | 0.26 | 0.24 | 0.26 | 0.17 | 0.24 |
| | Time | 0.34 | 0.33 | 0.35 | 0.35 | 0.34 |
| | SSUG | **0.48** | **0.43** | **0.40** | **0.62** | **0.40** |
| Nagoya | Distance | 0.25 | 0.21 | 0.26 | 0.12 | 0.24 |
| | Time | 0.31 | 0.29 | 0.35 | 0.27 | 0.31 |
| | SSUG | **0.34** | **0.43** | **0.41** | **0.67** | **0.36** |
| Fukuoka | Distance | 0.27 | 0.20 | 0.27 | 0.09 | 0.25 |
| | Time | 0.34 | 0.30 | 0.38 | 0.21 | 0.33 |
| | SSUG | **0.40** | **0.50** | **0.51** | **0.63** | **0.43** |
| Sendai | Distance | 0.26 | 0.21 | 0.26 | 0.08 | 0.24 |
| | Time | 0.32 | 0.28 | 0.35 | 0.20 | 0.31 |
| | SSUG | **0.34** | 0.29 | **0.39** | 0.32 | **0.34** |
| Sapporo | Distance | 0.27 | 0.13 | 0.23 | 0.05 | 0.25 |
| | Time | 0.34 | 0.21 | 0.33 | 0.16 | 0.33 |
| | SSUG | **0.42** | **0.54** | **0.49** | **0.72** | **0.44** |

Table notes: Os - Ds indicates suburban origin to suburban destination trips; Os - Du indicates suburban origin to urban destination trips; Ou - Os indicates urban origin to suburban destination trips; Ou - Ou indicates urban origin to urban destination trips.

The strength of the SSUG model lies in its ability to account for the heterogeneity in human mobility, particularly the differences between urban and suburban commuting patterns. Our results indicate that suburban areas tend to exhibit more diverse commuting behaviors, whereas urban areas have more centralized flows, concentrated around core business districts. This distinction enhances the SSUG model's performance, as it is better able to capture the varied commuting propensities in these regions. In Sendai, however, the relatively weaker central hub results in less pronounced differences between urban and suburban areas, probably leading to more modest improvements in the SSUG model's predictions. As Sendai has the smallest scale among the selected regions, future study hould examine whether cities with smaller scales than Sendai exhibit this feature. It is important to understand whether the issues in Sendai are tied specifically to its size or if they reflect broader trends in smaller metropolitan areas.

These findings highlight the importance of considering regional differences, particularly in sparsely populated areas. In metropolitan regions where urban cores play a dominant role in attracting commuters, the SSUG model proves to be a valuable tool for predicting and managing commuting flows. Understanding these dynamics is crucial for formulating effective urban policies, particularly those aimed at mitigating the adverse effects of urban sprawl [4,13], which depicts a scenario where high-density urban centers continue to attract resources and population, while suburban areas risk becoming increasingly isolated and less economically viable. It is essential to address these challenges through data-driven models like the SSUG to promote balanced urban development.

## 5 Conclusion

This study introduced the Spatially Segregated Urban Gravity (SSUG) model to enhance the accuracy of commuting flow predictions across six major Japanese metropolitan areas—Tokyo, Osaka, Nagoya, Fukuoka, Sendai, and Sapporo. By incorporating commuting time as a decay factor and distinguishing between urban and suburban commuting behaviors, the

SSUG model demonstrates superior predictive accuracy, particularly for urban-to-urban flows, compared to traditional gravity models.

Our key findings are threefold:

1. The SSUG model significantly improves upon traditional gravity models by incorporating commuting time and classifying commuting trips by type, with the most substantial enhancements observed in urban commuting flows.
2. While maintaining the core structure of the gravity model, the SSUG model reveals distinct spatial patterns between urban and suburban areas, confirming different commuting behaviors and contributing to a deeper understanding of urban polycentricity [20].
3. The SSUG model suggests that urban residents and employees often form a single cohesive group, offering new insights into urban center dynamics. It also reveals that urban and suburban regions follow different strengths of scaling laws in their commuting patterns.

This study addressed three primary questions:

1. What quantitative insights can be derived from human mobility models?
2. How can the gravity model serve as a tool for urban classification?
3. Under what conditions is the SSUG model most effective in predicting commuting flows?

Our results confirm that commuting time is a more effective decay factor than distance in predicting flows, particularly in densely populated metropolitan regions. The model's parameters reveal heterogeneity in commuting behaviors between urban and suburban areas, with the SSUG model performing better as the disparity between these regions increases. Furthermore, we demonstrate the gravity model's utility as an urban function classification tool, capable of distinguishing between urban and suburban commuting patterns.

A key strength of the SSUG model lies in its use of high-accuracy GPS data, which simplifies data collection and processing while maintaining strong predictive accuracy. This feature enhances the model's practicality for urban planners by reducing complexity and mitigating privacy concerns. Beyond transportation planning, the model's ability to capture the heterogeneity of commuting behaviors between urban and suburban regions offers valuable insights for policy-making, particularly in optimizing urban land use and reducing traffic congestion.

The SSUG model demonstrates strong potential for generalizability beyond Japan through two fundamental characteristics. First, the functional urban zones it identifies—primary urban cores ($h_u w_u$), residential-dominant areas ($h_u w_s$), employment centers ($h_s w_u$), and low-density periphery ($h_s w_s$)—represent universal metropolitan archetypes documented across global urban systems [41,42]. While spatial configurations vary, these functional roles in generating commuting flows are consistent across diverse metropolitan contexts.

Second, the methodology's transferability stems from its purely data-driven approach, requiring only origin-destination flow data without dependence on region-specific administrative boundaries or zoning classifications. With GPS, mobile network, or survey data increasingly available worldwide, researchers can apply this framework to identify local-specific parameters ($h_c$, $w_c$) and scaling relationships governing metropolitan mobility. The SSUG model thus provides a standardized yet adaptable framework for comparative urban analysis across diverse territorial realities.

However, the model's applicability to cities smaller than Sendai remains uncertain, necessitating further study to explore its effectiveness across different urban contexts. One possible reason is the choice of transportation mode; transportation patterns impact commuting behavior, as people's choice of residence and workplace locations often depends on their preferred mode of transport and associated commuting time tolerance. In Japanese metropolitan areas, the extensive public transit system plays a crucial role in shaping these choices, while smaller cities may show different patterns due to higher car dependency [44]. Applying the SSUG model to smaller cities or regions would provide valuable insights into its scalability and broader applicability across different urban contexts and transportation systems.

While the SSUG model demonstrates strong performance, several limitations should be acknowledged. First, our analysis relies on data from 2021, a period affected by both the COVID-19 pandemic and the Tokyo Olympics, which may have influenced typical commuting patterns. Future validation with data from more typical periods would help establish the model's robustness across different temporal contexts, and a long-term observation might help unravel spatial-temporal changes, such as major enterprise relocations or newly developed city centers. Additionally, other factors beyond urban and suburban segmentation—such as public transportation accessibility, human activity levels, and socioeconomic variables—may also influence commuting flows.

A further limitation of this study lies in the assumption that commuting time for an OD pair corresponds to the minimum observed travel time. While this approach is justified by the principle of travel time optimization and helps filter out non-commuting detours, it represents a simplification of real-world behavior. The model therefore does not fully account for the variability in travel times caused by different transportation modes, route choices, and real-time congestion. This assumption may impact the model's conclusions; for example, the overestimation of flows for certain trip types, such as from suburban origins to urban destinations, could be partially explained by this limitation. In these regions, a small number of individuals with access to faster private transport may define the minimum travel time, while the majority who rely on slower public transport are not fully represented. Future research could explore more complex deterrence functions that account for modal choice and travel time variability.

In conclusion, the SSUG model represents a significant advancement in understanding and predicting urban commuting patterns. By integrating spatial segregation into the traditional gravity model framework, it offers a more nuanced and accurate tool for urban planners and policymakers, contributing to the development of more efficient and sustainable metropolitan areas.

## Appendix A.  F-Test formulas and calculations

The F-test was used to compare the variances between the segregated and combined gravity models.

### Appendix A.1.  Hypotheses

- Null Hypothesis ($H_0$): No significant difference in variances between the segregated and combined models.
- Alternative Hypothesis ($H_1$): Significant difference in variances between the segregated and combined models.

### Appendix A.2.  F-Statistic calculation

The F-statistic is calculated using the formula:

$$F = \frac{MS_A}{MS_\epsilon} = \frac{(S_1 - S_2)/(k-1)}{S_2/\phi_{e2}} \tag{A1}$$

Where:

- $S_1$: This represents the sum of squared residuals (SSR) from the model assuming a common slope across all groups.
- $S_2$: This represents the sum of squared residuals (SSR) from the model allowing for different slopes for each group.

$MS_A$ (Mean Square for the slopes) represents the variance estimate due to the difference between the slopes of the regression lines for the different groups.

It is calculated as the difference in the sum of squared residuals between the model with a common slope ($S_1$) and the model with individual slopes ($S_2$), divided by the associated degrees of freedom.

Essentially, $MS_A$ helps to quantify how much the slopes differ from one another, which is then used in the F-test to determine if those differences are statistically significant.

$MS_\epsilon$ (Mean Square Error or Mean Square of the Residuals) represents the variance within groups due to the residuals (errors) in the regression model. It is calculated as the sum of squared residuals from the model allowing for individual slopes ($S_2$), divided by its associated degrees of freedom.

In other words, $MS_\epsilon$ quantifies the variability in the data that cannot be explained by the regression model, serving as a measure of the unexplained error within the groups. It is used as the denominator in the F-test to compare against $MS_A$.

### Appendix A.3.  Degrees of freedom calculation

**Appendix A.3.1.  Degrees of freedom for the common slope model.**  The degrees of freedom ($\phi_{e1}$) are calculated as the total degrees of freedom minus the degrees of freedom for the regression with a common slope.

**Appendix A.3.2.  Degrees of freedom for the separate slopes model.**  The degrees of freedom ($\phi_{e2}$) are calculated as the total degrees of freedom minus the degrees of freedom for the regression with separate slopes.

### Appendix A.4. Significance level

A significance level of 0.05 was used. The null hypothesis is rejected if the F-statistic exceeds the critical value from the F-distribution table at this significance level.

### Appendix A.5. Interpretation

**Appendix A.5.1. Why does $\frac{MS_A}{MS_\epsilon}$ (F-value) indicate the significance of individual slope regressions?** The F-value is used to test whether the variability explained by the differences in slopes among groups (represented by $MS_A$) is significantly greater than the variability due to random error within the groups (represented by $MS_\epsilon$). If the F-value is large, it suggests that the slopes are significantly different from each other, indicating that the individual slope regressions explain a meaningful amount of variance beyond what would be expected by chance.

**Appendix A.5.2. Does the F-value calculated follow an F-distribution?** The F-value, calculated as $\frac{MS_A}{MS_\epsilon}$, follows an F-distribution under the null hypothesis that all group slopes are equal. The degrees of freedom used in the F-distribution depend on the degrees of freedom associated with $MS_A$ and $MS_\epsilon$. This distribution allows us to determine the significance of the F-value and thus assess the hypothesis regarding the slopes' equality.

**Appendix A.5.3. Process.** To rigorously test this hypothesis, we first conducted a preliminary statistical analysis to determine whether dividing the dataset into two populations based on numerically defined population scale thresholds results in significantly different gravity model parameters.

- **Baseline specification.** We first estimate the conventional log-linear gravity model,

$$\log F_{ij} = \alpha + \beta_1 \log H_i + \beta_2 \log W_j + \gamma \log t_{ij} + \varepsilon_{ij},$$

  by ordinary least-squares (OLS).
- **F-test on population splits.** Because the model is linear in logs, OLS residuals allow a classical F-test (ANOVA) to check whether the coefficient vector differs across population-density groups.
- **Scanning all candidate thresholds.** We sweep every integer value of the residential and workplace population cut-offs $(h, w)$ within the observed range (see Fig 1). For each split we (i) re-estimate the log-linear model in the two sub-samples and (ii) compute the F-statistic against the pooled model. The assumptions of the test—normality and homoscedasticity of residuals hold throug centain ranges.
- **Rationale for Bayesian optimisation (BO).** Once the F-test confirms that *some* statistically significant segmentation exists, we embed the cut-offs as continuous parameters $(h_c, w_c)$ in the nonlinear SSUG formulation and let Bayesian optimisation jointly estimate $(h_c, w_c)$ together with the eight segment-specific coefficients, yielding fully data-driven thresholds.

## Appendix B. Detailed Bayesian optimization method

### Appendix B.1. Introduction

Bayesian optimization is a powerful method for optimizing complex, noisy, and expensive-to-evaluate objective functions. It is particularly useful for hyper parameter tuning in machine learning and other computational models. In this study, we applied Bayesian optimization to determine the eight parameters for the gravity model.

### Appendix B.2. Bayesian optimization framework

The Bayesian optimization process can be summarized in the following steps:

1. Initialization:
   - Begin with a small initial set of observations (e.g., threshold values and corresponding objective function evaluations).
2. Surrogate Model:
   - Construct a probabilistic model of the objective function. Common choices include Gaussian Processes (GPs), Random Forests, and Tree-structured Parzen Estimators (TPE). In this study, we used Gaussian Processes due to their flexibility and well-calibrated uncertainty estimates.
3. Acquisition Function:
   - Define an acquisition function that quantifies the utility of evaluating the objective function at a given point. Common acquisition functions include Expected Improvement (EI), Probability of Improvement (PI), and Upper Confidence Bound (UCB). We used the Expected Improvement function, which balances exploration and exploitation by selecting points with the highest expected improvement over the current best observation.
4. Optimization of Acquisition Function:
   - Optimize the acquisition function to select the next point to evaluate. This step involves solving a simpler optimization problem to find the point that maximizes the acquisition function.
5. Evaluate Objective Function:
   - Evaluate the true objective function at the selected point and update the surrogate model with this new observation.
6. Iterate:
   - Repeat steps 3-5 until a stopping criterion is met (e.g., a maximum number of iterations or convergence threshold).

### Appendix B.3. Application to gravity model

In our study, the objective function to be optimized was the fit of the gravity model, measured by the objective function written in Method session. The steps were as follows:

1. Initialization:
   - Randomly select an initial set of $\theta$ values and evaluate the gravity model's performance for these thresholds.
2. Surrogate Model:
   - Use a Gaussian Process to model the objective function. The GP provides a mean function and a covariance function to predict the objective function value at any point in the threshold space.
3. Acquisition Function:
   - Calculate the Expected Improvement (EI) for potential $\theta$ values.
4. Optimization of Acquisition Function:
   - Use a gradient-based optimizer to find the $\theta$ values that maximize the EI.
5. Evaluate Objective Function:
   - Compute the objective function for the selected $\theta$ values by evaluating the gravity model's performance.

6. Iteration:
  - Repeat the optimization and evaluation steps until convergence.

## Supporting information

**S1 File. This file contains all supplementary information.** The contents are as follows: (1) Study area; (2) Commuter trajectory; (3) Commuting distance and time; (4) Deviation of the gravity model; (5) Prediction of the SSUG model; (6) Commuting propensities; (7) Regional analysis; (8) Robustness analysis; (9) Data bias; (10) CES form of the gravity model; (11) Clarification of Constraint 2 to Eq 4; (12) OD flow and commuting drivers; (13) Objective function values; (14) Clearer subplots for Fig 4.
(PDF)

## Author contributions

**Conceptualization:** Yixuan Y. Zheng.

**Data curation:** Yixuan Y. Zheng.

**Formal analysis:** Yixuan Y. Zheng.

**Funding acquisition:** Misako Takayasu.

**Investigation:** Yixuan Y. Zheng.

**Methodology:** Yixuan Y. Zheng, Hideki Takayasu.

**Project administration:** Misako Takayasu.

**Resources:** Misako Takayasu.

**Supervision:** Misako Takayasu.

**Validation:** Yixuan Y. Zheng, Yohei Shida, Hideki Takayasu.

**Visualization:** Yixuan Y. Zheng.

**Writing – original draft:** Yixuan Y. Zheng.

**Writing – review & editing:** Yixuan Y. Zheng, Yohei Shida, Hideki Takayasu, Misako Takayasu.

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
