## [Decision Letter · Decision Letter 0]

27 Jan 2025

PONE-D-24-49532Enhancing the gravity model for commuters: Time-and-spatial-structure-based improvements in Japan’s metropolitan areasPLOS ONE

Dear Dr. Zheng,

Thank you for submitting your manuscript to PLOS ONE. After careful consideration, we feel that it has merit but does not fully meet PLOS ONE’s publication criteria as it currently stands. Therefore, we invite you to submit a revised version of the manuscript that addresses the points raised during the review process. I apologize for the delay handling your manuscript. The two qualified reviewers had opposite recommendation, but both provided useful suggestions. I also read your paper carefully and lean to the second reviewer who recommended a minor revision. The first reviewer commented on the typesetting of your figures or images; this may not be a problem when the paper is published online, but you need to check the visual quality of your image files. I myself have one comment. A common problem faced by commuters is road congestion. Neither distance alone nor commute time alone can reflect the degree of congestion, but both do--the ratio of distance to time, or speed. I am not sure if you should or can incorporate this to your gravity model, but you may at least discuss this. Perhaps in urban areas in Japan, most commuters use public transit? For the reviewer comments, please try to respond as many as you can or provide a rebuttal.

We look forward to receiving your revised manuscript.

Kind regards,

Shihe Fu, Ph.D.

Academic Editor

PLOS ONE

3. Thank you for stating the following financial disclosure:  [Grant-in-Aid for Scientific Research (B) (Grant Number 22H01711)].  Please state what role the funders took in the study.  If the funders had no role, please state: "The funders had no role in study design, data collection and analysis, decision to publish, or preparation of the manuscript." If this statement is not correct you must amend it as needed.

4. For studies involving third-party data, we encourage authors to share any data specific to their analyses that they can legally distribute. PLOS recognizes, however, that authors may be using third-party data they do not have the rights to share. When third-party data cannot be publicly shared, authors must provide all information necessary for interested researchers to apply to gain access to the data. (https://journals.plos.org/plosone/s/data-availability#loc-acceptable-data-access-restrictions)

6. We note that Figure 5 and 6 in your submission contain [map/satellite] images which may be copyrighted. All PLOS content is published under the Creative Commons Attribution License (CC BY 4.0), which means that the manuscript, images, and Supporting Information files will be freely available online, and any third party is permitted to access, download, copy, distribute, and use these materials in any way, even commercially, with proper attribution. For these reasons, we cannot publish previously copyrighted maps or satellite images created using proprietary data, such as Google software (Google Maps, Street View, and Earth). For more information, see our copyright guidelines: http://journals.plos.org/plosone/s/licenses-and-copyright.

1. You may seek permission from the original copyright holder of Figure 5 and 6  to publish the content specifically under the CC BY 4.0 license. 

Additional Editor Comments (if provided):

Reviewers' comments:

Reviewer's Responses to Questions

**Comments to the Author**

1. Is the manuscript technically sound, and do the data support the conclusions?

Reviewer #1: Partly

Reviewer #2: Yes

2. Has the statistical analysis been performed appropriately and rigorously? 

Reviewer #1: No

Reviewer #2: Yes

3. Have the authors made all data underlying the findings in their manuscript fully available?

Reviewer #1: No

Reviewer #2: Yes

4. Is the manuscript presented in an intelligible fashion and written in standard English?

Reviewer #1: Yes

Reviewer #2: Yes

5. Review Comments to the Author

Reviewer #1: The manuscript still suffers a lot in research design and analysis.

SSUG is not well highlighted in the discussions.

The parameters are estimated merely using OLS. Please see the attached file with detailed comments.

Reviewer #2: This paper makes a clear contribution in extending the gravity model to predicting commuting flows within cities. Specifically, the paper allows the scaling parameters between urban areas and suburban areas to be different to accommodate potential non-linearity in the traditional log-log linear gravity model. Beyond that, the paper spends efforts in differentiating working population and residential population at the same location, with high quality GPS data. By doing so, the paper effectively allows asymmetric commuting between urban/suburban residential locations and urban/suburban working locations, thereby enhancing the traditional gravity model to a large extent.

I just have a few comments for the authors:

1. The paper identifies working/residential locations with some criterion. This could potentially lead to a problem: Users in the GPS data could have multiple working/residential locations. If that’s true, how do the authors calculate commuting time?

2. The paper uses data in 2021, which seems to be an unusual year with both covid pandemic and Tokyo Olympics. I worry that the results might not have external validity in predicting commuting flows in other years and other regions.

3. Could the SSUG model be more flexible in terms of functional form? Currently the model features the traditional gravity form with log-log linearity. If the paper tries the more general CES form, the model could well deliver even better performance.

6. PLOS authors have the option to publish the peer review history of their article (what does this mean?). If published, this will include your full peer review and any attached files.

Reviewer #1: No

Reviewer #2: No

---

## [Author Response · Author response to Decision Letter 1]

3 Mar 2025

We sincerely thank the editor for taking the time to carefully review our manuscript and provide valuable comments. We deeply appreciate the thorough reviews from both reviewers, as their detailed feedback has helped us identify areas where we can improve clarity and strengthen our methodology.

Since there are tables and figures contained in the response letter, please check the attached Response to Reviewers.pdf for details.

---

## [Decision Letter · Decision Letter 1]

27 May 2025

PONE-D-24-49532R1Enhancing the gravity model for commuters: Time-and-spatial-structure-based improvements in Japan’s metropolitan areasPLOS ONE

Dear Dr. Zheng,

Thank you for submitting your manuscript to PLOS ONE. After careful consideration, we feel that it has merit but does not fully meet PLOS ONE’s publication criteria as it currently stands. Therefore, we invite you to submit a revised version of the manuscript that addresses the points raised during the review process.

We look forward to receiving your revised manuscript.

Kind regards,

Qing-Chang Lu

Academic Editor

PLOS ONE

**Additional Editor Comments:**

While recognizing the values of this work, there are still some comments to be addressed.

Reviewers' comments:

Reviewer's Responses to Questions

**Comments to the Author**

1. If the authors have adequately addressed your comments raised in a previous round of review and you feel that this manuscript is now acceptable for publication, you may indicate that here to bypass the “Comments to the Author” section, enter your conflict of interest statement in the “Confidential to Editor” section, and submit your "Accept" recommendation.

Reviewer #2: All comments have been addressed

Reviewer #3: All comments have been addressed

Reviewer #4: (No Response)

2. Is the manuscript technically sound, and do the data support the conclusions?

Reviewer #2: (No Response)

Reviewer #3: (No Response)

Reviewer #4: Partly

3. Has the statistical analysis been performed appropriately and rigorously? 

Reviewer #2: (No Response)

Reviewer #3: Yes

Reviewer #4: N/A

4. Have the authors made all data underlying the findings in their manuscript fully available?

Reviewer #2: (No Response)

Reviewer #3: Yes

Reviewer #4: Yes

5. Is the manuscript presented in an intelligible fashion and written in standard English?

Reviewer #2: (No Response)

Reviewer #3: Yes

Reviewer #4: Yes

6. Review Comments to the Author

Reviewer #2: (No Response)

Reviewer #3: thank you for allowing me to comment on this interesting article.

The article proposes a new form of the classical gravity model of regional studies, corrected by urban structures. This innovation is novel and combined with a very interesting empirical validation.

In that sense, I would like to suggest a broader explanation of the assumptions of the methodology and how these assumptions limit the conclusions (specifically the assumption that in the model individuals always take the fastest route).

On the other hand, is it possible to complexify the urban segmentation sub irbans by variables beyond population change?

In terms of form, it would be interesting to improve the quality of the illustrations and equations to make them more readable as well as to improve the conclusions, exploring how the conclusions can be transferred to territorial realities other than Japan.

Reviewer #4: This manuscript presents the Spatially Segregated Urban Gravity (SSUG) model, which addresses key challenges in accurately predicting metropolitan commuting flows. The model’s novel approach of accounting for urban and suburban commuting dynamics, along with the use of Bayesian optimization, is compelling and relevant for urban mobility studies. In my review, I raised several technical issues that require clarification, such as the rationale for specific constraints, the real-world meaning of mixed-type grids, and the robustness of commuting time measures. Additionally, I highlighted concerns about the validity of using the F-test in this context. Overall, this manuscript is a valuable contribution and I believe that addressing these comments will help to strengthen its methodological rigor and ensure the findings are robust and well-substantiated.

7. PLOS authors have the option to publish the peer review history of their article (what does this mean?). If published, this will include your full peer review and any attached files.

Reviewer #2: No

Reviewer #3: **Yes: **Carlos Aguirre-Nuñez

Reviewer #4: No

---

## [Author Response · Author response to Decision Letter 2]

23 Jun 2025

We sincerely thank the editor and all reviewers for their invaluable feedback that has significantly strengthened our manuscript.

To the Editor: We are deeply grateful for your time, patience, and commitment throughout this review process. Your willingness to continue working with our manuscript, particularly given that it had been reviewed previously, means a great deal to us and demonstrates exceptional editorial dedication. Your careful coordination of the review process have been instrumental in bringing this work to its current improved state.

To Reviewer #3: Your constructive suggestions have greatly enhanced the readability and applicability of our paper. Your guidance on clarifying methodology assumptions and limitations has been particularly beneficial, providing clear structural direction for our study. Your emphasis on making assumptions and their corresponding limitations explicit first has improved our research framework significantly. Additionally, your focus on transferability highlighted the key value of our study for future development in this research field.

To Reviewer #4: Your detailed theoretical suggestions have substantially enhanced the credibility of our work from a fundamental perspective. Your comprehensive comments on assumptions, constraint settings, and result comparisons were especially valuable in strengthening our methodological rigor. Most importantly, your request for clearer explanation of the huws and hswu area classifications helped us better articulate this key component of our paper, which forms the core innovation of our approach.

The collective feedback from all reviewers has resulted in a much stronger, clearer, and more impactful manuscript. We believe the revisions have significantly improved both the theoretical rigor and practical applicability of our research.

Thank you again for your dedication to advancing scholarly research in this field.

---

## [Decision Letter · Decision Letter 2]

14 Jul 2025

PONE-D-24-49532R2Enhancing the gravity model for commuters: Time-and-spatial-structure-based improvements in Japan’s metropolitan areasPLOS ONE

Dear Dr. Zheng,

Thank you for submitting your manuscript to PLOS ONE. After careful consideration, we feel that it has merit but does not fully meet PLOS ONE’s publication criteria as it currently stands. Therefore, we invite you to submit a revised version of the manuscript that addresses the points raised during the review process.

We look forward to receiving your revised manuscript.

Kind regards,

Qing-Chang Lu

Academic Editor

PLOS ONE

Journal Requirements:

Additional Editor Comments:

I would like to invite the authors to address one further comment.

Reviewers' comments:

Reviewer's Responses to Questions

**Comments to the Author**

1. If the authors have adequately addressed your comments raised in a previous round of review and you feel that this manuscript is now acceptable for publication, you may indicate that here to bypass the “Comments to the Author” section, enter your conflict of interest statement in the “Confidential to Editor” section, and submit your "Accept" recommendation.

Reviewer #3: All comments have been addressed

Reviewer #4: (No Response)

2. Is the manuscript technically sound, and do the data support the conclusions?

Reviewer #3: Yes

Reviewer #4: Partly

3. Has the statistical analysis been performed appropriately and rigorously? 

Reviewer #3: Yes

Reviewer #4: Yes

4. Have the authors made all data underlying the findings in their manuscript fully available?

Reviewer #3: Yes

Reviewer #4: Yes

5. Is the manuscript presented in an intelligible fashion and written in standard English?

Reviewer #3: Yes

Reviewer #4: Yes

6. Review Comments to the Author

Reviewer #3: Thank you for taking on board my suggestions, from my perspective the article is now clearer, I suggest its publication.

Reviewer #4: Thank you for your thoughtful and thorough revision. I appreciate the authors' efforts to address the issues raised in the first round of review. In my judgment, all of the concerns have been adequately addressed except for Comment 1, regarding Constraint 1.

In my detailed response (attached), I explain why the current justification for Constraint 1 remains problematic unless the authors clarify the strong exclusivity assumptions it implies for the population and employment data. Specifically, the condition that fij = 1 when hi = 1 and wj = 1 leads to logical inconsistencies if more than one zone has one resident or one job. I recommend that the authors explicitly acknowledge these implications and revise the manuscript accordingly. Please refer to the attached file for detailed comments.

Apart from this point, I find the revisions to be well reasoned and appropriate, and I appreciate the authors' careful attention to the previous review.

7. PLOS authors have the option to publish the peer review history of their article (what does this mean?). If published, this will include your full peer review and any attached files.

Reviewer #3: **Yes: **Carlos Aguirre

Reviewer #4: No

---

## [Author Response · Author response to Decision Letter 3]

15 Jul 2025

Dear Editor and Reviewers,

We are grateful for the opportunity to submit a second revision of our manuscript and sincerely

appreciate the constructive feedback provided during the review process. The additional comments

have been invaluable in helping us strengthen the methodological rigor and clarity of our work.

To the Editor: We extend our sincere appreciation for your expert management of the review

process and for facilitating the constructive dialogue between authors and reviewers. We are grateful

for your patience and professional oversight during the revision process.

To Reviewer 4: We thank you for your diligent and detailed second review. We agree that your

remaining comment regarding the logical consistency of Constraint 1 is a critical point. In response,

we have focused this revision on addressing this issue directly. As detailed below, we have clarified

the data preprocessing rule that ensures the validity of this constraint, thereby resolving the potential

inconsistency you identified.

To Reviewer 3: We extend our heartfelt thanks for your positive assessment and recommendation

for publication. Your supportive evaluation of our manuscript is greatly appreciated.

We believe these final revisions substantially enhance the manuscript’s methodological rigor and

address the remaining concerns raised during the review process. The specific changes and detailed

explanations are outlined in the attached Response_to_Reviewers.pdf.

---

## [Editor Report · Decision Letter 3]

21 Jul 2025

Enhancing the gravity model for commuters: Time-and-spatial-structure-based improvements in Japan’s metropolitan areas

PONE-D-24-49532R3

Dear Dr. Zheng,

We’re pleased to inform you that your manuscript has been judged scientifically suitable for publication and will be formally accepted for publication once it meets all outstanding technical requirements.

Kind regards,

Qing-Chang Lu

Academic Editor

PLOS ONE
---

## [Editor Report · Acceptance letter]

PONE-D-24-49532R3

PLOS ONE

Dear Dr. Zheng,

I'm pleased to inform you that your manuscript has been deemed suitable for publication in PLOS ONE. Congratulations! Your manuscript is now being handed over to our production team.

Kind regards,

on behalf of

Dr. Qing-Chang Lu

Academic Editor

PLOS ONE